# Delivery of a post-natal neonatal jaundice education intervention improves knowledge among mothers at Jinja Regional Referral Hospital in Uganda

**Businge Alinaitwe** [1¤*], **Nkunzimaana Francis**[2], **Tom Denis Ngabirano**[1], **Charles Kato**[3], **Petranilla Nakamya**[4], **Rachel Uwimbabazi**[5], **Adam Kaplan**[6], **Molly McCoy**[7], **Elizabeth Ayebare**[1], **Jameel Winter** [8]

1 Department of Nursing, College of Health Sciences, Makerere University, Kampala, Uganda, 2 Clinical Epidemiology Unit, School of Medicine, College of Health Sciences, Makerere University, Kampala, Uganda, 3 School of Medicine, College of Health Sciences, Makerere University, Kampala, Uganda, 4 School of Public Health, College of Health Sciences, Makerere University, Kampala, Uganda, 5 Public Health Commission, Boston, Massachusetts, United States of America, 6 Center for Care Delivery and Outcomes Research, Minneapolis, Minnesota, United States of America, 7 Global Programs and Strategy Alliance, University of Minnesota, Minneapolis, Minnesota, United States of America, 8 Department of Neonatology, Children's Minnesota, Minneapolis, Minnesota, United States of America

¤ Current address: School of Nursing, Mount Kenya University, Thika, Kenya
* busingebruceali@gmail.com

**Data Availability Statement:** The data supporting the findings of this study are publicly available from

## Abstract

### Background

Neonatal jaundice (NNJ) is a major contributor to childhood morbidity and mortality. As many infants are discharged by 24 hours of age, mothers are key in detecting severe forms of jaundice. Mothers with limited knowledge of NNJ have a hard time identifying these infants who could go on to have the worst outcomes. This study aimed to determine the effect of a jaundice education package delivered to mothers prior to hospital discharge on maternal knowledge after discharge.

### Methods

This was a before and after interventional study involving an education package delivered through a video message and informational voucher. At 10–14 days after discharge, participants were followed up via telephone to assess their post-intervention knowledge. A paired t-test was used to determine the effectiveness of the intervention on knowledge improvement. Linear regression was used to determine predictors of baseline knowledge and of change in knowledge score.

### Results

Of the 250 mothers recruited, 188 were fit for analysis. The mean knowledge score was 10.02 before and 14.61 after the intervention, a significant difference (p<0.001). Factors determining higher baseline knowledge included attendance of 4 or more antenatal visits (p

the figshare repository (https://doi.org/10.6084/m9.figshare.25348426).

**Funding:** Funding was provided by the University of Minnesota Center for Global Health and Social Responsibility through; A global research training under the Uganda Research Training Collaborative awarded to students from Makerere University and the University of Minnesota And then through the Center for Global Health and Social Responsibility's Global Engagement Grants program to support global health collaborations awarded to Dr. Jameel Winter as the Principal Investigator and Businge Alinaitwe as the Co-Principal Investigator. Website URL: https://globalhealthcenter.umn.edu/ The funder did not play any role in the study design, data collection and analysis, decision to publish, or preparation of the manuscript.

**Competing interests:** The authors have declared that no competing interests exist.

< 0.001), having heard about NNJ previously (p < 0.001), having experienced an antepartum illness (p = 0.019) and higher maternal age (p = 0.015). Participants with poor baseline knowledge (β = 7.523) and moderate baseline knowledge (β = 3.114) had much more to gain from the intervention relative to those with high baseline knowledge (p < 0.001).

## Conclusion

Maternal knowledge of jaundice can be increased using a simple educational intervention, especially in settings where the burden of detection often falls on the mother. Further study is needed to determine the impact of this intervention on care seeking and infant outcomes.

## Introduction

Globally, there has been substantial progress in decreasing child mortality, but greater effort is still needed in reducing neonatal mortality [1]. Neonatal mortality accounts for an increasing portion of the total under-five mortality [1, 2], with most recent data suggesting that 45% of under-five mortality occurs in the first 28 days of life.

Improving infant survival in the first month of life is a strategic step towards reducing under-5 mortality and will be paramount in achieving Sustainable Development Goal 3 by the year 2030 [3]. Unfortunately, the neonatal mortality rate in Uganda remains high at 19 deaths per 1,000 live births [2] with substantial regional variation. A study in Eastern Uganda demonstrated a mortality rate as high as 34 deaths per 1,000 live births, much higher than the nationally observed rate [4].

As a commonly diagnosed condition especially within the first week of life, severe neonatal jaundice is a major contributor to hospital admissions in the African setting as indicated by a systematic review by Slusher and colleagues [5]. About 1.1 million neonates develop severe neonatal jaundice (NNJ) in Sub-Saharan Africa and Southern Asia every year [6] and in 2016, severe NNJ was attributable to 1,309 deaths per 100,000 live births, ranking 7th among all causes of mortality in the first week of life [7].

Although over 60% of all newborns develop some degree of jaundice, NNJ is often times a self-limiting process. However, a significant portion of newborn babies end up developing severe NNJ as indicated in Kampala Uganda where about one quarter of infants diagnosed with NNJ developed severe hyperbilirubinemia [8]. Severe jaundice if left untreated, can be a significant contributor to neonatal morbidity and mortality [9, 10]. Morbidities such as delay in reaching development milestones, motor delays including cerebral palsy, visual impairment and auditory deficits, seizure or death can also occur due to bilirubin encephalopathy [11].

In Eastern Uganda, newborns who develop one or more danger sign(s) have an increased risk of dying in the first 28 days of life [4]. Severe neonatal jaundice (NNJ) is a common diagnosis at Jinja Regional Referral Hospital in Eastern Uganda, leading to significant healthcare costs [12].

However, it is common for new mothers to have limited understanding of these danger signs during the postpartum period in Ugandan rural communities [13]. In addition, neonatal jaundice is one of the most poorly recognized and least known danger sign by mothers [14–16]. Given that about one half of all infants born in health facilities in Uganda are discharged home before 24 hours of life [17]. Often times severe jaundice occurring after hospital discharge may go undiagnosed if parents are unaware of the signs and symptoms to watch for [18].

After discharge from the hospital, babies are under their mother's direct supervision most of the time. Therefore, if well informed, mothers are vital in the early detection of neonatal jaundice and seeking for appropriate and timely care [19], thus avoiding exposure to harmful traditional practices which could otherwise exacerbate neonatal jaundice [20, 21].

In Uganda, there is limited literature on maternal neonatal jaundice knowledge and no study has tested the effectiveness of an educational package in improving maternal knowledge of neonatal jaundice. This study therefore aimed at determining the effect on maternal knowledge of an enhanced NNJ education package delivered to women in the post-delivery period before hospital discharge. The study also aimed to assess for other factors associated with maternal knowledge.

## Methods

### Study design and setting

This was a facility-based educational interventional study with a before and after design. Quantitative techniques of data collection were used in both the pre- and post-intervention assessments. The study took place in the postnatal ward of Jinja Regional Referral Hospital (JRRH) located in Jinja city, south-eastern Uganda. The hospital is a referral center for the districts Busoga sub-region and part of Central Uganda serving a population of approximately 4.5 million people with about 6000 births annually.

### Study population

The study population included all women who gave birth at JRRH during the time of the study. Participants were recruited while in the postpartum recovery ward to ensure that they received information on neonatal jaundice before they were discharged home as all of the women in the immediate postpartum are discharged through this ward. As neonatal jaundice commonly occurs in the first week of life [22], we delivered our intervention with in the first week post-delivery with three in four of all the mothers (73.4%) receiving the enhanced neonatal jaundice education intervention within 24 hours of giving birth. All women with live born infants including preterm and low birth weight babies were eligible for inclusion. Women who had an intrauterine fetal death and those in an unstable state postpartum were excluded. Additionally, patients who were referred to JRRH after delivery and those who had come back for post-natal check-up were excluded from the study.

### Sample size determination and sampling procedure

The sample size was calculated to attain a power of 80%, a two-sided probability of type one error of 5% and an outcome standard deviation of 0.3. One hundred ninety-five women were required to detect a mean percentage difference in the knowledge score of 20% between the pre-and post-knowledge assessments. We anticipated a loss to follow up of 20% and thus we increased our sample size by 20% to accommodate for this, resulting in a sample size of 234 participants.

Convenience sampling was used to recruit all participants while the study team was available. Participants were enrolled consecutively until the targeted sample size was obtained.

### Study variables

Maternal knowledge of NNJ was the outcome/dependent variable. The independent variables assessed included age, occupation, marital status, education level, residence, parity, antenatal

care attendance, gestation age at delivery, mode of delivery, the neonate's birth weight and sex as well as maternal clinical characteristics.

## Operational definitions

Poor knowledge: All participants that scored 0–9 knowledge points were categorized as having poor knowledge of neonatal jaundice.

Moderate knowledge: Participants that scored 10–14 knowledge points out of 25 were defined as having moderate knowledge.

Adequate knowledge: Participants that score >14 knowledge points out of 25 were categorized as having adequate knowledge of neonatal jaundice.

## Data collection instrument and quality control

Data were collected using structured questionnaires administered via face-to-face interview during the baseline assessment and via telephone interview for the follow-up assessment. The questionnaire was prepared in English, reviewed by a neonatologist and Midwifery specialists, and then translated into Lusoga, the most used language in the region. Before the actual data collection, the tool was pre-tested among 10 postnatal mothers at a health center in Eastern Uganda. The knowledge assessment tool was found to have a good internal consistency with a Cronbach's alpha of 0.81. The research assistants were trained on the process of data collection and implementation of the intervention.

## Data collection procedures

Data collection started on the 22nd day of December 2021 and ended on the 3rd day of February 2022. Data were collected from each participant on two occasions, before and after the educational intervention. Maternal neonatal jaundice knowledge was assessed using 12 jaundice knowledge questions measured on a scale of 0–25 knowledge points (S1 File).

Seven questions had one possible correct response and therefore the participant could obtain up to 7 knowledge points on these questions. Each of the remaining five questions had multiple correct responses, meaning the participant could score more than one knowledge point per question. The post-intervention survey was done 10–14 days after discharge and was completed via phone call (Fig 1).

Data from both the pre- and post-intervention surveys were entered directly into an offline Kobo Toolbox database (Boston, MA, USA). Research assistants (RAs) checked the information for accuracy and completeness and data was then synchronized by the RAs to a server accessible only to the research team.

## Intervention and delivery strategy

The enhanced neonatal jaundice educational intervention was a maternal tailored information package that presented the definition of NNJ, a general overview of newborn danger signs, signs of NNJ, risk factors and causes of NNJ, how to check for NNJ, complications associated with severe untreated NNJ, general maternal care for infants with jaundice, care seeking for severe NNJ and other danger signs as well as hospital-based treatment modalities for severe NNJ. The intervention was developed from the National Institute for Health and Clinical Excellence (NICE) NNJ guidelines [23]. It was aimed to improve maternal knowledge of neonatal jaundice. The intervention was provided to all study participants before discharge. It was delivered to each participant individually in a private space. This setup allowed for physical distancing to prevent COVID-19 transmission. For mothers that were delivered by cesarean

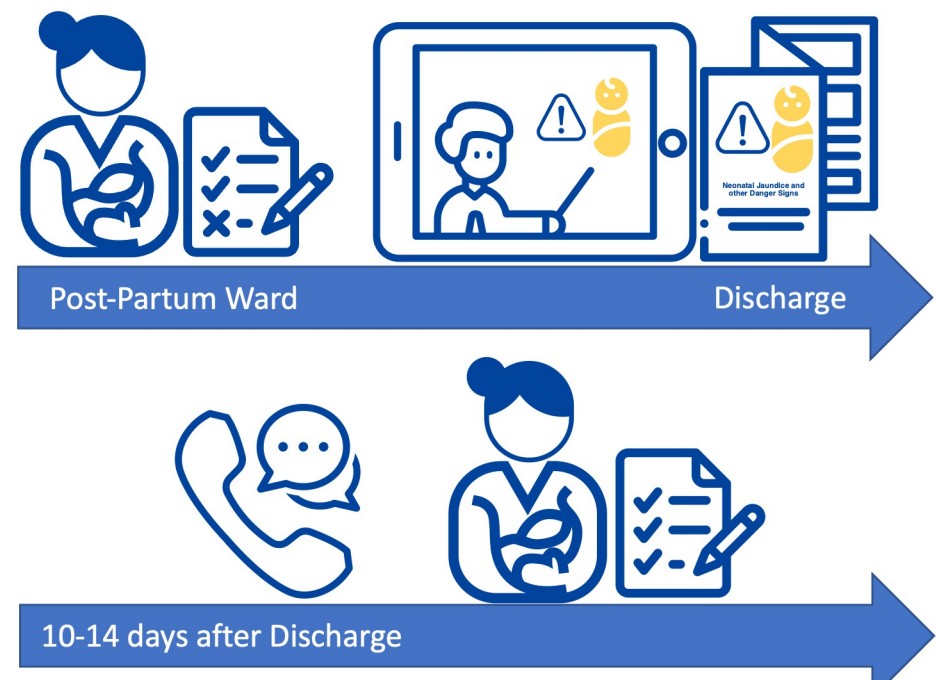

**Fig 1. Enhanced neonatal jaundice education study flow diagram.**

section and unable to ambulate, the intervention was received from their recovery bed on the ward. The NNJ messages were given in three steps (Fig 1):

1. After the baseline assessment, the study Midwives gave a general overview of NNJ to each participant.

2. A video presentation narrated in either Lusoga or English was presented to participants which they watched on a 7-inch tablet in the presence of the research midwife. The Lusoga and English videos were 10 and 5 minutes in length respectively. Upon completion of the video, the mothers were given an opportunity to ask questions.

3. Mothers were provided with informational leaflets containing the same information as that in the video. The leaflets were printed in color to provide clear pictures for a jaundiced baby and were also available in both Lusoga and English. Participants were allowed to take these leaflets to their homes to always read and familiarize themselves with neonatal jaundice.

## Statistical measures & data management

Descriptive statistics including mean, and standard deviation were performed for all continuous variables as well as frequencies and percentages for categorical variables. Results were presented in tables. The effect of the intervention on maternal knowledge was evaluated based on the average change in the mother's knowledge after the intervention. A paired sample t-test statistic was used to assess the change in knowledge score. Stepwise linear regression analysis was used to establish the factors that affected both pre-intervention knowledge score and the change in mothers' knowledge score. P-values, 95% confidence intervals and the R-Square

values were calculated. All analyses were done using Stata Version 17.0/MP and R statistical software (R Core Team, 2022).

## Ethical considerations

The study was approved by the Makerere University School of Health Sciences Research and Ethics Committee, (approval number MUSSS-2020-10), University of Minnesota Institutional Review Board and Research and Ethics Committee (STUDY00011629), and the Uganda National Council of Science and Technology (UNCST). Participants were informed about the study to ensure that they were able to make an informed decision on whether to participate or not. Written informed consent was obtained from every participant that agreed to participate in the study. The patient signed to acknowledge consent and this was followed by the research assistant's signature. Participant identifying information was stored in locked cabinets with keys accessible to only the study team. Emancipated minors were consented following the same process for consenting adults.

## Results

### Socio-demographic characteristics

In this study, a total of 265 women were screened but 250(94.3%) met the eligibility criteria and were enrolled. During the post-intervention telephone follow-up, 189 (75.6%) were successfully contacted and therefore included in the final analysis. However, the post-intervention knowledge score was missing for one participant (analytical data set N = 188). Two thirds (67.6%) of our participants reside in Jinja district and one third in other districts of Central and Eastern Uganda. The mean age of study participants was 25.0 years (SD ±4.99) with a range of 16 to 39 years. Majority (92.0%) of the participants were aged 18 to 34 years and 2.7% were below 18 years (Table 1).

### Obstetric, clinical and neonatal characteristics

More than one half (55.9%) of the participants were multiparous, most (91.5%) had a term delivery, and the highest parity was 8. The mean number of antenatal care visits was 4.77 (SD ±1.49) and the mean birth weight was 3.15 kg (SD ±0.56) with the smallest baby weighing 1.3 kg and the largest being 5.0 kg. Most of the babies (90.4%) had a normal birth weight ($\geq$2.50 kilograms) (Table 1).

### Maternal neonatal jaundice knowledge

Most (83.6%) of the participants had some prior awareness of neonatal jaundice. Only 2.1% reported having a previous child with NNJ. In the pre-intervention assessment, the mean knowledge score was 10.02 (SD ±3.95) points (Fig 2). The minimum score was 0 and the maximum score was 18. One half (50.0%) of women had moderate knowledge, 40.4% had poor knowledge and only 9.6% had adequate knowledge of neonatal jaundice.

After the intervention, the average knowledge score increased to 14.61 points (Fig 2) (SD ±2.31), with a minimum of 7 knowledge points and a maximum of 21 knowledge points. More than half (53.2%) had adequate knowledge, several (44.7%) had moderate knowledge and only 2.1% had poor knowledge. The highest change in knowledge after the intervention was observed to occur among women from the moderate to the high knowledge category at 40.0% followed by a change from poor to adequate category at 23.9% (Table 2). A paired t-test demonstrated a significant increase in post-intervention mean knowledge scores by 4.59 points (p<0.00).

**Table 1. Demographic, obstetric, clinical and new-born characteristics of study participants.**

| Variable | Absolute frequency (n = 188) | Relative frequency (%) |
|---|---|---|
| **Residence** | | |
| Jinja | 127 | 67.6 |
| Elsewhere | 61 | 32.3 |
| **Age. mean = 25.0 SD±4.99** | | |
| <18 years | 05 | 2.7 |
| 18–34 years | 173 | 92.0 |
| ≥35 years | 10 | 5.3 |
| **Education level** | | |
| No formal schooling | 14 | 7.4 |
| Completed primary | 30 | 16.0 |
| Completed secondary | 97 | 51.6 |
| Completed tertiary | 47 | 24.9 |
| **Marital status** | | |
| Single | 14 | 7.4 |
| Married | 174 | 92.5 |
| **Occupation** | | |
| House wife | 96 | 51.1 |
| Self-employed | 57 | 30.3 |
| Employed | 35 | 18.6 |
| **Parity (mean = 2.14, SD±1.34)** | | |
| Primiparous | 83 | 44.1 |
| Multiparous | 105 | 55.9 |
| **ANC attendance** | | |
| Yes | 185 | 98.4 |
| No | 3 | 1.6 |
| **ANC no. of times (N = 186), Mean = 4.77, SD±1.60** | | |
| <4 | 30 | 16.0 |
| ≥4 | 158 | 84.0 |
| **Hypertension** | | |
| Yes | 17 | 9.0 |
| No | 171 | 91.0 |
| **HIV Status** | | |
| Negative | 180 | 95.7 |
| Positive | 8 | 4.3 |
| **Antepartum illness(es)** | | |
| Yes | 89 | 47.3 |
| No | 99 | 52.7 |
| **Gestation age at delivery; Mean = 38.00, SD ±1.58** | | |
| ≤36 weeks | 16 | 8.5 |
| ≥37 weeks | 172 | 91.5 |
| **Mode of delivery** | | |
| Vaginal | 130 | 69.1 |
| Cesarean-Section | 58 | 30.9 |
| **Sex of the baby** | | |
| Male | 81 | 43.1 |
| Female | 107 | 56.9 |
| **Birthweight; Mean 3.15, SD±0.56** | | |

*(Continued)*

**Table 1.** (Continued)

| Variable | Absolute frequency (n = 188) | Relative frequency (%) |
|---|---|---|
| Low birth weight (<2.5 kg) | 18 | 9.6 |
| Normal weight (≥2.5. kg) | 170 | 90.4 |
| **Breastfed (within 1 hour of life)** | | |
| Yes | 143 | 76.1 |
| No | 45 | 23.9 |
| **Reason for not breastfeeding (n = 45)** | | |
| Newborn related causes | 21 | 46.7 |
| Maternal related causes | 24 | 53.3 |
| **Ever heard about NNJ** | | |
| Yes | 157 | 83.5 |
| No | 31 | 16.5 |
| Time of NNJ intervention delivery | | |
| Within 24 hours | 138 | 73.4 |
| 24–48 hours | 12 | 6.4 |
| ≥48 hours | 38 | 20.2 |

## Predictors of pre-intervention NNJ maternal knowledge

Maternal socio-demographic, clinical and obstetric features were used as independent variables to determine factors associated with baseline NNJ knowledge score. In a stepwise linear regression model, age of the mother was one of the factors significantly associated with pre-intervention maternal knowledge with an increase in one year resulting in an increase in maternal knowledge score by 0.13 points (p = 0.015). In addition, attending 4 or more antenatal care visits, being free of antenatal illnesses, and having heard about neonatal jaundice before were significantly associated with a higher baseline NNJ knowledge score (Table 3).

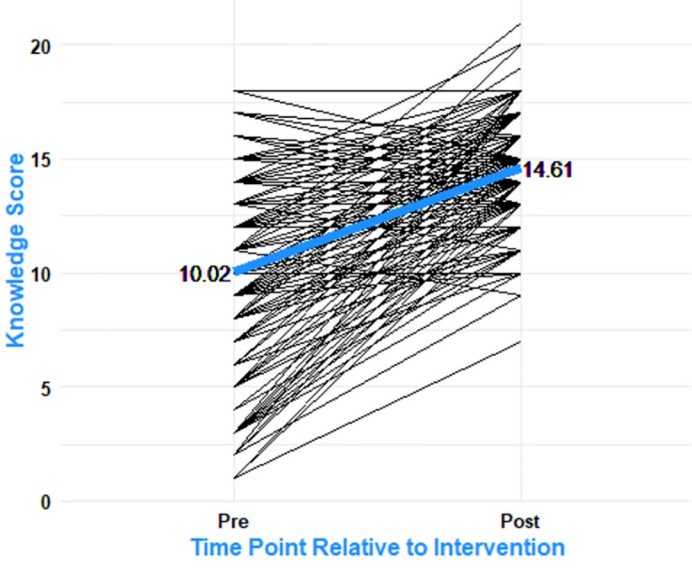

**Fig 2. Change in NNJ knowledge score before and after the intervention.** This represents the overall change in maternal NNJ knowledge score from pre- to post-intervention. The bold blue line shows the mean change, and the individual black lines represent the individual mothers' change in score.

**Table 2. Change in maternal knowledge between the poor, moderate and high categories.**

| Change in knowledge between groups | Frequency | Percentage (n = 188) |
|---|---|---|
| Remained in the poor group | 03 | 1.6 |
| Poor to moderate | 45 | 23.9 |
| Poor to adequate | 28 | 14.9 |
| Remained in moderate group | 35 | 18.6 |
| Moderate to poor | 01 | 0.5 |
| Moderate to adequate | 58 | 40.0 |
| Remained in adequate group | 14 | 7.4 |
| Adequate to moderate | 04 | 2.1 |
| Adequate to poor | 00 | 00 |
| Total | 188 | 100% |

## Predictors of change in NNJ knowledge score

We observed that those with poor knowledge before the intervention had much more to gain from the intervention (β = 7.720 and p-value < 0.001); likewise, those with moderate knowledge of jaundice had much more to gain from the intervention (β = 3.408 and p-value < 0.001) as compared to the adequate baseline knowledge group (Table 4). The R-squared value from this model was 0.52 indicating that this model explained roughly 52% of the variability in the mothers' change in knowledge scores. However, this percent-variability reflects that purely baseline knowledge, and not maternal and demographic features, drives knowledge gain.

## Discussion

In this study, we implemented an intervention in which we taught mothers about neonatal jaundice as a danger sign of a newborn baby. Two knowledge assessments, one before and the other after the intervention were performed.

In our study, most of the participants (83.5%) had previously heard about neonatal jaundice. Our findings are similar to studies conducted in Ghana [24–26] and Nigeria [21, 27, 28] an indicator of high level of previous exposure to neonatal jaundice information. However, one study from Effutu county in Ghana indicated that only 17.3% reported to have previous

**Table 3. Factors associated with pre-intervention maternal NNJ knowledge score at JRRH.**

| Variable | Coefficient | P-value | 95% CI |
|---|---|---|---|
| Increase in mother's age in years | 0.13 | 0.015 | 0.03–0 .22 |
| Ever heard about NNJ | 4.01 | <0.001 | 2.67–5.34 |
| Attending 4 or more ANC visits | 2.65 | <0.001 | 1.29–3.99 |
| Suffering from antepartum illness(s) | -1.18 | 0.019 | -2.16–0.19 |

**Table 4. Factors associated with change in maternal NNJ knowledge score.**

| Variable | Coefficient | P-value | 95% CI |
|---|---|---|---|
| Never Heard of Jaundice | -1.189 | 0.034 | -2.285–0.092 |
| Baseline knowledge score—Poor (relative to adequate) | 7.523 | < 0.001 | 6.123–8.918 |
| Baseline knowledge score—Moderate (relative to adequate) | 3.114 | < 0.001 | 21.777–4.451 |

awareness of neonatal jaundice [29], which is very low when compared to our findings. This difference could be due to high literacy levels observed in our study, as 76.5% of the participants had completed secondary education and above when compared to 66.6% in Ghana.

When compared to the pre-intervention knowledge level in our study, evidence elsewhere demonstrates that the level of maternal knowledge of NNJ in Uganda is in dire need of improvement, and the enhanced NNJ education package was implemented at the right moment. For instance, whereas in Egypt, Nepal, Ethiopia and Nigeria, adequate knowledge scores were found at 52.3% [30], 22% [31], 39.2% [32] and 57.1% [28] respectively, in our study, only 9.6% of mothers in the baseline assessment demonstrated adequate knowledge of NNJ.

Adequate levels of NNJ in our study were only comparable to the situation in Egypt and Nigeria after the enhanced NNJ education intervention [28, 30]. This variation could be due to the possibility of having NNJ knowledge improvement strategies already present as routine practice in these countries. Additionally, this result may be a reflection of our assessment tool being different from other assessment tools, or the difference in scoring and grading of knowledge scores.

The enhanced NNJ educational intervention implemented in our study was effective in improving maternal knowledge of NNJ after discharge from hospital, with a statistically significant increase in mean knowledge scores between the pre- and post-intervention knowledge assessments. This improvement in knowledge score reflects an increase in the number of mothers able to identify NNJ as a danger sign of a newborn baby, identify the causes, signs and symptoms, where to check for NNJ and general care for a baby with NNJ. This is clinically significant as it reflects increased maternal awareness of the problem. An intervention study conducted by Kashaki and colleagues in Iran demonstrated similar findings with maternal involvement coined as a pivotal step for proper management of neonatal jaundice [33].

The improvement in NNJ knowledge after an intervention may be attributed to a favorable individualized learning environment where every woman received the education in a one-on-one session with the research Midwife. Additionally, the use of the video and information vouchers served as visual learning aids to enhance knowledge assimilation, which were different from the conventional group-based health education. The one-on-one individualized interactions between the Midwife and the mother during our intervention provided an opportunity to ensure adequate information sharing and comprehension of the messages on NNJ.

The current study asserts the notion that maternal health education improves awareness on health challenges with the possibility of increasing patient engagement in self-care. In Eastern Uganda, a study focusing on reducing neonatal mortality demonstrated that maternal involvement in newborn and infant care was key towards achieving better infant outcomes [34].

It is worth noting that the intervention was able to improve the knowledge beyond the levels that were previously observed in Nepal and Ethiopia [31, 32]. Simple interventions as this are therefore vital if Uganda, a country in Sub Saharan Africa where neonatal mortality contributes to 55% of all global deaths is to be compared with countries like Egypt in North Africa where neonatal mortality is only at 4% of the global share [35].

Age was found to be a significant predictor of pre-intervention maternal knowledge of NNJ. This finding was expected as with age comes increased exposure and awareness on the common challenges of a newborn baby. This is similar to findings from Ghana among antenatal and postnatal mothers [36]. Since young mothers are more likely to be inexperienced with low knowledge levels, healthcare providers need to initiate and promote targeted health education sessions to cater for these vulnerable mothers and their infants.

Antenatal care visits were found to be strongly associated with neonatal jaundice knowledge. Attendance of 4 or more ANC visits was positively associated with high maternal NNJ

knowledge. Moreover, for every additional antenatal care visit attended, there was an associated increase in the NNJ knowledge score by 2.65 points. This is not surprising because during ANC, mothers receive guidance regarding pregnancy, birth preparedness and care of the infant after birth. They would therefore be expected to be well informed regarding newborn health. Similar to our findings, Demis and colleagues in Northern Ethiopia demonstrated that antenatal care follow-up was a significant predictor of maternal NNJ knowledge [32, 33] and overall, ANC has been found to improve maternal knowledge for indicators of newborn illness [37, 38].

The strong association between adequate antenatal care and neonatal jaundice knowledge highlights the importance of ANC for the infant's future wellbeing, and care providers should emphasize the key neonatal jaundice messages during these vital sessions to promote the mother's pre-delivery knowledge. This is very important as our study found out that mothers who had prior awareness of NNJ significantly got higher pre-intervention knowledge scores. Salia and colleagues in Ghana also found that prior exposure to neonatal jaundice information was a significant predictor of maternal knowledge [25]. Antenatal care is such a key entry point for promoting continuum of care and its significance in new-born wellbeing has been illustrated by different studies with observed significant reduction in neonatal mortality [39, 40].

Given that the majority (98.4%) of our participants were able to attend one or more antenatal care visits and 84.0% received ≥4 ANC visits, most of our participants had the opportunity of exposure to neonatal jaundice messages prenatally. The result from our study indicates a higher level of ANC than findings from rural Ghana where only 69% of participants attended 4+ ANC visits [36]. The difference could be due to the low education levels. However, the low NNJ knowledge levels observed in our study regardless of the high antenatal care attendance levels warrants further investigation.

The generalizability of our findings in the Ugandan setting could be challenging as majority of the women (84.0%) in our study had attended 4 or more ANC visits, 76.5% had an education level of secondary and above, 100% delivered from a health facility and therefore received the immediate postnatal care. The knowledge levels observed before the intervention may therefore not be entirely representative of Ugandan women.

In a study by McDiehl and colleagues in rural Uganda, only 71% of mothers completed at least 4 or more ANC visits [41]. In a nationally representative study, this level was even lower at 59% [42, 43]. Given that about 60% of women deliver from a health facility and only 65% receive postnatal care in the immediate postpartum period [44], there is need to assess women who may not be represented in our study and create an educational package that takes into account of the contextual differences. More so, increasing ANC attendance, providing the education package in ANC and increasing the level of health facility delivery could be vital in ensuring that women receive this type of intervention at the point of care.

## Limitations

During the follow-up assessment, it was difficult to contact mothers whose phones were either off or when the phone owners were not at home by the time the team contacted them. This was mitigated by calling several times as well as calling in the evening hours when owners of phones had come back from work.

The study involved a pre and post intervention knowledge assessment but there was no comparison group. The change in knowledge after the intervention cannot therefore be explained with certainty by the education intervention. Our assessment tool was not a validated assessment device, which may have confounded some of our baseline knowledge levels,

however it still demonstrated an improvement in NNJ knowledge. Additionally, the pre and post knowledge assessments were conducted in two different environments, the hospital and home settings respectively, and this could have contributed to variations in responses to the knowledge questions. During the phone call, we encouraged the patient to find a quiet distraction-free environment.

Our study design leant itself to simple analytic methods which may have overlooked some confounders such as the following. With our frequencies per district, we were not exactly powered to detect more detailed regional differences, that is, we only investigated Jinja versus all other districts. As to be expected, neonatal jaundice knowledge and awareness are not uniformly present throughout Uganda. In future work, a promising feature to collect would be distance between home and hospital since access to healthcare may impact awareness of neonatal jaundice.

## Conclusion

As jaundice remains one of the leading causes of neonatal morbidity and mortality worldwide, there is a need for simple interventions like this to improve maternal knowledge of this potentially devastating condition, especially in settings where the burden of detection often falls on the mother. In this study, we demonstrated that maternal knowledge of jaundice can be improved using a simple educational intervention prior to discharge. The strong association between adequate antenatal care and neonatal jaundice knowledge highlights the importance of ANC for the infant's future wellbeing. Health care providers should utilize the antenatal period to enhance mother's knowledge on NNJ. Further study is required to determine impact of such an intervention on detection of severe NNJ, care seeking and infant outcomes. More so, women who deliver from home and those who do not receive the first post-delivery care need to be considered in future research studies.

## Supporting information

**S1 File. Neonatal jaundice (NNJ) knowledge assessment questions.**
(DOCX)

## Acknowledgments

We appreciate Faith Kisakye, Faith Niwasaasira, and Kagoya Viola for their support during data collection. We acknowledge University of Minnesota Center for Global Health and Social Responsibility (CGHSR) for providing this research mentorship program, Makerere University for fostering the collaboration with University of Minnesota CGHSR as well as Jinja Regional Referral Hospital for accommodating this study. We also specifically thank all the participants that took part in this study.

## Author Contributions

**Conceptualization:** Businge Alinaitwe, Nkunzimaana Francis, Tom Denis Ngabirano, Charles Kato, Petranilla Nakamya, Rachel Uwimbabazi, Molly McCoy, Elizabeth Ayebare, Jameel Winter.

**Data curation:** Nkunzimaana Francis, Adam Kaplan.

**Formal analysis:** Nkunzimaana Francis, Tom Denis Ngabirano, Petranilla Nakamya, Adam Kaplan, Elizabeth Ayebare.

**Investigation:** Businge Alinaitwe, Tom Denis Ngabirano, Charles Kato, Elizabeth Ayebare, Jameel Winter.

**Methodology:** Businge Alinaitwe, Nkunzimaana Francis, Tom Denis Ngabirano, Charles Kato, Petranilla Nakamya, Rachel Uwimbabazi, Adam Kaplan, Elizabeth Ayebare, Jameel Winter.

**Project administration:** Businge Alinaitwe, Tom Denis Ngabirano, Molly McCoy.

**Resources:** Tom Denis Ngabirano, Molly McCoy.

**Supervision:** Businge Alinaitwe, Tom Denis Ngabirano, Charles Kato, Adam Kaplan, Elizabeth Ayebare, Jameel Winter.

**Validation:** Adam Kaplan, Elizabeth Ayebare, Jameel Winter.

**Visualization:** Adam Kaplan, Jameel Winter.

**Writing – original draft:** Businge Alinaitwe, Nkunzimaana Francis, Tom Denis Ngabirano, Charles Kato, Petranilla Nakamya, Rachel Uwimbabazi, Adam Kaplan, Molly McCoy, Elizabeth Ayebare, Jameel Winter.

**Writing – review & editing:** Businge Alinaitwe, Nkunzimaana Francis, Tom Denis Ngabirano, Charles Kato, Petranilla Nakamya, Rachel Uwimbabazi, Adam Kaplan, Molly McCoy, Elizabeth Ayebare, Jameel Winter.

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
