## [Decision Letter · Decision Letter 0]

21 Dec 2023

PONE-D-23-35071Delivery of a post-natal neonatal jaundice education intervention improves knowledge among mothers at Jinja Regional Referral Hospital in UgandaPLOS ONE

Dear Dr. Alinaitwe,

Thank you for submitting your manuscript to PLOS ONE. After careful consideration, we feel that it has merit but does not fully meet PLOS ONE’s publication criteria as it currently stands. Therefore, we invite you to submit a revised version of the manuscript that addresses the points raised during the review process.

We look forward to receiving your revised manuscript.

Kind regards,

Germana Bancone, Ph.D

Academic Editor

PLOS ONE

Journal Requirements:

"We appreciate Faith Kisakye, Faith Niwasaasira, and Kagoya Viola for their support during data collection. 

We acknowledge University of Minnesota Center for Global Health and Social Responsibility for 

financially supporting this study, Makerere University for fostering the collaboration with University of 

Minnesota and Jinja Regional Referral Hospital for accommodating this study. We also specifically thank 

all the participants that took part in this study."

"Funding was provided by the University of Minnesota Center for Global Health and Social Responsibility through;

A global research training under the Uganda Research Training Collaborative awarded to students from Makerere University and the University of Minnesota

And then through the Center for Global Health and Social Responsibility’s Global Engagement Grants program to support global health collaborations awarded to Dr. Jameel Winter as the Principal Investigator and Businge Alinaitwe as the Co-Principal Investigator.

Website URL: https://globalhealthcenter.umn.edu/

The funder did not play any role in the study design, data collection and analysis, decision to publish, or preparation of the manuscript."

Reviewers' comments:

Reviewer's Responses to Questions

**Comments to the Author**

1. Is the manuscript technically sound, and do the data support the conclusions?

Reviewer #1: Yes

Reviewer #2: Partly

2. Has the statistical analysis been performed appropriately and rigorously? 

Reviewer #1: Yes

Reviewer #2: I Don't Know

3. Have the authors made all data underlying the findings in their manuscript fully available?

Reviewer #1: Yes

Reviewer #2: Yes

4. Is the manuscript presented in an intelligible fashion and written in standard English?

Reviewer #1: Yes

Reviewer #2: Yes

5. Review Comments to the Author

Reviewer #1: Thank you for giving me the opportunity to review your work.

The article describes an educational intervention for women in the period immediately after giving birth which aims to increase the level of awareness of neonatal jaundice. The burden of neonatal jaundice remains largely underestimated in Africa and there is little work on women’s education, therefore, this article adds something new to the state of the art.

The manuscript is very well written and informative. The intervention was well structured and although it is a pilot study, it is important in my view that these results are published as they can inform larger studies designed in such a way as to overcome some of the limitations present in this first design (limitations well described by the authors themselves), as well as for studying whether this intervention has a practical impact.

Minor points:

- Some information is repeated several times in the text and/or scattered (e.g. consent) and perhaps the manuscript could be revised to avoid this problem and make it more concise.

- Tables: In my view some data could be merged, like Table 1 and 2 for instance, to have 1 table describing the study cohort more concisely.

One question, are the obstetric data self-reported or taken from hospital records? Or both?

Other very minor points:

Line 57 ref [5] is quite specific, while the data presented are quite common, a more generic reference (like a review) would be more appropriate.

Line 68: I might see where you are going with the sentence, but I suggest to rephrase it to convey better the message.

Reviewer #2: Thank you for allowing me to review this manuscript on the evaluation of a neonatal jaundice education package offered to Ugandan mothers before discharge from the post-natal ward.

Please find below my comments:

Introduction:

The introduction is quite detailed however its content is sometimes redundant and doesn’t allow the text to flow smoothly.

Methods:

Line 101: quantitative and qualitative techniques of data collection were used: from the suppl. File and the results, we only see quantitative data – was the qualitative part done?

Study population: could the authors clarify a little more the rational of doing an educational intervention at the postpartum ward exclusively? And at what time in the postpartum? In the introduction and also in the discussion, it is mentioned that most women go back home at 24h post-delivery. It would be therefore important to know when this intervention was done.

Sample size: I am not quite sure I understand why the sample size was calculated to detect a percentage difference in knowledge – as there is only one intervention and one population tested the difference in knowledge was expected for which factor? And based on what information? In addition, what do the authors mean by “an outcome standard deviation of 0.3”?

S1 knowledge tool: question 32 should have 2 correct responses if I understand the questionnaire correctly. In addition, I am unsure of the procedure: as reported presently it looks like for all the questions but the 2 mentioned as “do not give the mother answers” (22 and 32), the interviewers were listing the possible answers including for questions 27 for example?

Could the authors clarify why they 1. Decided that some questions would be answered with help and some without and 2. Why the interviewers had to conduct the interview rather than letting the mother respond to the questionnaire herself considering the high education level of all these women?

Could the authors also explain why they chose to recall the knowledge within 10-14 days after the return home (line 131) or is it 7-14 days (line 170)?

Study variables: this part contains analytic methods to move to the appropriate section. Line 131 is similar to line 151. As in the introduction, the flow of explanations is not smooth, there is some information on the knowledge tool Line 129 (similar to Line 159), cut-offs in “operational definitions” and details on the questionnaire itself and the scoring Lines 155 onwards – it would be better if all the information was under a same heading.

Operational definitions: is this scoring as per NICE guidelines or decided by the authors?

Intervention and delivery strategy: again, a lot of redundancy in this part of the text with what was said before – please revise the flow of your methods.

Statistical measures: I am not sure I understand what was the linear measure used? Was it the change in knowledge? Which results include the multi-predictor analysis and the stepwise linear regression analysis? Tables 3 and 4 seems to be univariate analysis?

Results:

Table 2 shows a large number of women with c-sections – when did the authors interview these women? Were they fit to receive that kind of information?

I am surprised the authors did not compare the characteristics of multiparous and nulliparous. This is actually a point mentioned in the discussion and it would have been interesting to see its confirmation in the results.

Maternal neonatal jaundice knowledge: “Most of the participants scored between 8 to 15 points” – however this range of numbers is outside the 3 operational definitions – maybe remove and give also the absolute numbers for each of those 3 categories instead?

If the mean change in score is 4 points then it is still within moderate knowledge (from 10 to 14). It would be interesting to evaluate how many mothers went from poor to moderate or adequate knowledge and from moderate to adequate. It would be also interesting to evaluate what happened to the women who decreased their score – any potential explanation for this?

Predictors of maternal knowledge: Line 269 – I suggest rephrasing the sentence to avoid using the term “change in knowledge” which may lead to confusion as this is also one of the outcomes measured.

Table 3: the variable “no formal schooling” – was it evaluated as a dichotomic variable (no formal schooling yes/no)?

Discussion:

The first sentence (Line 284) should be in the introduction I think to help understand better the burden of NNJ in this setting. Lines 287-288 are a bit misleading as the references for this sentence are from Ethiopia – presenting as it is now, it looks like this is a part of the results of this study.

Line 301-303: I would agree that the knowledge has overall improved, but how “significant” was it from a point of view of caring for neonates? The authors mentioned in the methods that one part of the questionnaire was on the well-being of the child and the mother at the 2nd interview – do the authors know if mothers brought their child to check the jaundice afterward? I would also be interested to understand why the environment was favorable (the immediate post-delivery period may not be the best environment for concentrating on new knowledge or is it?).

Line 325: women in this study have high education level (line 296) and most have heard about neonatal jaundice (Line 294), so I am a little surprised to read that the better adequate knowledge obtained at baseline in other settings could be due to higher education levels and living in an urban setting – could there be another explanation? Were the questionnaires the same? Is the scoring identical?

Overall, there is a point the authors should discuss a little further: the population recruited had a high education level, went to the minimal requirement of ANC consultations, was mostly employed, had heard from neonatal jaundice, and delivered in a tertiary hospital. Are those results representative of a more general population? Would the general knowledge on neonatal jaundice be even lower? Would all the pregnant women be receptive to this type of information (i.e. videos and leaflets)? What would be the recommendations of the authors to move forward?

6. PLOS authors have the option to publish the peer review history of their article (what does this mean?). If published, this will include your full peer review and any attached files.

Reviewer #1: No

Reviewer #2: No

---

## [Author Response · Author response to Decision Letter 0]

6 Mar 2024

We thank you for the productive review and feedback provided to us. All responses to comments have been provided. We look forward to hearing from you

---

## [Decision Letter · Decision Letter 1]

18 Mar 2024

Delivery of a post-natal neonatal jaundice education intervention improves knowledge among mothers at Jinja Regional Referral Hospital in Uganda

PONE-D-23-35071R1

Dear Dr. Alinaitwe,

We’re pleased to inform you that your manuscript has been judged scientifically suitable for publication and will be formally accepted for publication once it meets all outstanding technical requirements.

Kind regards,

Germana Bancone, Ph.D

Academic Editor

PLOS ONE

Additional Editor Comments (optional):

Reviewers' comments:

Reviewer's Responses to Questions

**Comments to the Author**

1. If the authors have adequately addressed your comments raised in a previous round of review and you feel that this manuscript is now acceptable for publication, you may indicate that here to bypass the “Comments to the Author” section, enter your conflict of interest statement in the “Confidential to Editor” section, and submit your "Accept" recommendation.

Reviewer #2: All comments have been addressed

2. Is the manuscript technically sound, and do the data support the conclusions?

Reviewer #2: (No Response)

3. Has the statistical analysis been performed appropriately and rigorously? 

Reviewer #2: (No Response)

4. Have the authors made all data underlying the findings in their manuscript fully available?

Reviewer #2: (No Response)

5. Is the manuscript presented in an intelligible fashion and written in standard English?

Reviewer #2: (No Response)

6. Review Comments to the Author

Reviewer #2: (No Response)

7. PLOS authors have the option to publish the peer review history of their article (what does this mean?). If published, this will include your full peer review and any attached files.

Reviewer #2: No

---

## [Editor Report · Acceptance letter]

26 Mar 2024

PONE-D-23-35071R1 

PLOS ONE

Dear Dr. Alinaitwe, 

I'm pleased to inform you that your manuscript has been deemed suitable for publication in PLOS ONE. Congratulations! Your manuscript is now being handed over to our production team.

Kind regards, 

on behalf of

Dr. Germana Bancone 

Academic Editor

PLOS ONE